# Orthogonal control of mean and variability of endogenous genes in a human cell line

Alain R. Bonny[1,4], João Pedro Fonseca [1,3,4], Jesslyn E. Park[1] & Hana El-Samad [1,2✉]

Stochastic fluctuations at the transcriptional level contribute to isogenic cell-to-cell heterogeneity in mammalian cell populations. However, we still have no clear understanding of the repercussions of this heterogeneity, given the lack of tools to independently control mean expression and variability of a gene. Here, we engineer a synthetic circuit to modulate mean expression and heterogeneity of transgenes and endogenous human genes. The circuit, a Tunable Noise Rheostat (TuNR), consists of a transcriptional cascade of two inducible transcriptional activators, where the output mean and variance can be modulated by two orthogonal small molecule inputs. In this fashion, different combinations of the inputs can achieve the same mean but with different population variability. With TuNR, we achieve low basal expression, over 1000-fold expression of a transgene product, and up to 7-fold induction of the endogenous gene *NGFR*. Importantly, for the same mean expression level, we are able to establish varying degrees of heterogeneity in expression within an isogenic population, thereby decoupling gene expression noise from its mean. TuNR is therefore a modular tool that can be used in mammalian cells to enable direct interrogation of the implications of cell-to-cell variability.

[1] Department of Biochemistry and Biophysics, University of California, San Francisco, San Francisco, CA 94158, USA. [2] Chan Zuckerberg Biohub, San Francisco, CA 94158, USA. [3] Present address: Amyris Bio Products Portugal, Porto, Portugal. [4] These authors contributed equally: Alain R. Bonny, João Pedro Fonseca. ✉email: Hana.El-Samad@ucsf.edu

Clonal cells within a population can display tremendous variability in their physical characteristics (e.g., morphology), their molecular contents, as well as their transcriptional and signaling states, leading to distinct phenotypes and cell fate decisions in response to the same stimulus[1–6]. The roots of this cell-to-cell variability within a population are many, including distinct microenvironments or paracrine signaling[7]. However, a main driver of population heterogeneity is at the level of transcription[8–10]. Although certain "housekeeping" genes (e.g., ribosome biogenesis) can display remarkably uniform expression between cells, other genes can exhibit widely heterogeneous expression[11]. Uncovering how transcriptional fluctuations differentially direct downstream network activities to cause divergent behaviors from the average of the population has been a long-standing research effort[12,13]. In unicellular organisms, cell-to-cell variability in gene expression has been shown to confer survival under extreme duress in a phenomenon known as bet-hedging[14,15]. Bet-hedging in microbial populations is one example where variable transcriptional activity can drive phenotypic behaviors, which has been implicated in antibiotic resistance[16–18]. In multicellular organisms, transcriptional heterogeneity has been observed to at least partially influence cell fate decisions such as stem cell differentiation[13] and the HIV latent-active decision[19]. Furthermore, heterogeneity has been extensively documented as the potential underlying cause of drug resistance upon selection, a phenomenon that is influenced by the steady-state distribution of phenotypic states upon drug administration[20,21].

Although numerous observations implicating gene expression heterogeneity in differential phenotypes have been documented, determining the causal effect of this variability can only be done when it is the only experimental variable that is changed in a study. This has proven to be challenging because genetic manipulations that change variability, e.g., through the suppression or overexpression of a gene, also change mean gene expression. Interrogating hypotheses about variability therefore awaits strategies that can deconvolve the effects of changes in the mean expression of a gene from its variability. A synthetic biology approach is uniquely suited to address this challenge[20,22–27], as shown through the use of optogenetic pulsing[28], negative and positive feedback[20], as well as titratable, independent production and degradation of a protein of interest[29–31]. While presenting valuable proofs of concepts, these strategies remain challenging to deploy for biological studies in mammalian systems. For example, the optogenetic pulsing strategy allows for the same circuit to overexpress and independently modulate gene variability in response to different inputs of blue light. However, this strategy has only been vetted for transgene regulation in *Saccharomyces cerevisiae*, with non-trivial barriers to implementation in mammalian systems with the inherent complexity that arises in moving to a system in higher eukaryotes[32,33]. The strategy relying on the use of negative and positive feedback to regulate the mean and heterogeneity in gene expression requires different genetic circuits and cell lines, to achieve similar means with different variances, representing a cell engineering challenge. Lastly, although controlling protein production independent of its degradation is an elegant implementation to modulate mean and variability, this circuit relies on inserting a transgene and appending a destabilizing domain to the protein of interest, potentially perturbing its endogenous function. In addition, modifying endogenous loci with the destabilization domain is not modular, nor does it allow for high-throughput testing of different genes. Although each of the aforementioned studies has advanced our understanding, a strategy that is amenable to a wide range of mammalian expression systems, is modular to target transgene and endogenous loci, and can decouple changes in mean from variance, is still needed. To address this challenge, we

looked to an earlier synthetic circuit that utilized a serial orientation of independent inducible transcription factors to decouple mean expression from variability[34].

In this work, we engineer an analogous small molecule dual-inducible synthetic circuit in human cells, which we name a Tunable Noise Rheostat (TuNR), to independently titrate the steady-state mean expression and variability of transgene and endogenous gene products in a mammalian cell line (PC9). This cascading-activator circuit arrangement achieves ~1000-fold induction of transgene expression. Furthermore, different dosage regimes of the two small molecules could achieve the same mean gene expression (isomeans) with different variability within a population. As a proof-of-concept, we deployed TuNR to the endogenous loci of genes *NGFR* and *CXCR4*. Used in this endogenous context, the circuit can induce expression up to 7.2-fold for *NGFR* and 3.4-fold for *CXCR4*. In both cases, however, we could achieve isomean combinations of inducers where TuNR can modulate the variability of *NGFR* and *CXCR4* expression independent of their means. These data position TuNR as a modular circuit that allows protein mean expression and variability control, enabling systematic explorations of the specific consequences of mean expression of a gene versus its variability in mammalian cells.

## Results

### Characterization of a serial circuit topology with two inducible transcriptional activators.
We built TuNR as a serial connection of two inducible transcriptional activation systems, where the upstream system (first node) controls production of the downstream system (second node) (Fig. 1A and Supplementary Fig. 1A). The first node consists of a Gal4 DNA-binding domain fused to half of a split abscisic acid (ABA)-binding domain, which, in the presence of ABA, assembles with its cognate heterodimer fused to a VP-16 activation domain[35,36]. The recruitment of the ABA-reconstituted gene product of the first node to the upstream activating sequence minimal promoter drives the expression of the second inducible system and an mRuby as a reporter for transcription at this node of the cascade. The second node consists of a *Staphylococcus pyogenes* nuclease-dead Cas9 (dCas9) N-terminally fused to half of a gibberellic acid (GA)-binding domain and a VPR (p65, VP65, Rta) activation domain appended to the other half of the GA binding domain. In the presence of GA, these two proteins dimerize and, upon the concomitant expression of a target guide RNA (gRNA), are able to induce expression of the gene of interest (Fig. 1A). We identified ABA and GA as small molecule inducers of choice due to their previous vetting in other mammalian systems, reversibility of cognate protein dimerization, and the independence of each heterodimerization event[35–37]. Moreover, we chose dCas9 as the final node of TuNR for its modularity in targeting any locus with an appropriate protospacer adjacent motif.

We integrated TuNR together with a gRNA cassette targeting the Tetracycline Response Element (pTRE) and a pTRE-driven mAzamiGreen reporter in PC9 cells (Fig. 1A). To limit confounding effects of random integration of the circuit, we isolated and propagated cell lines from single-cell clones. To characterize the steady-state expression of the first node of the circuit, we induced expression with varying concentrations of ABA and measured mRuby expression daily over 7 days, replenishing the induction media every 24 h. We observed graded mRuby activation at all doses with each reaching steady state by day 3 and maintaining their respective mRuby expression for the remainder of the experiment (Fig. 1B and Supplementary Fig. 1C). Next, we characterized expression of the second node in a separate experiment by maximally priming cells with 400 μM

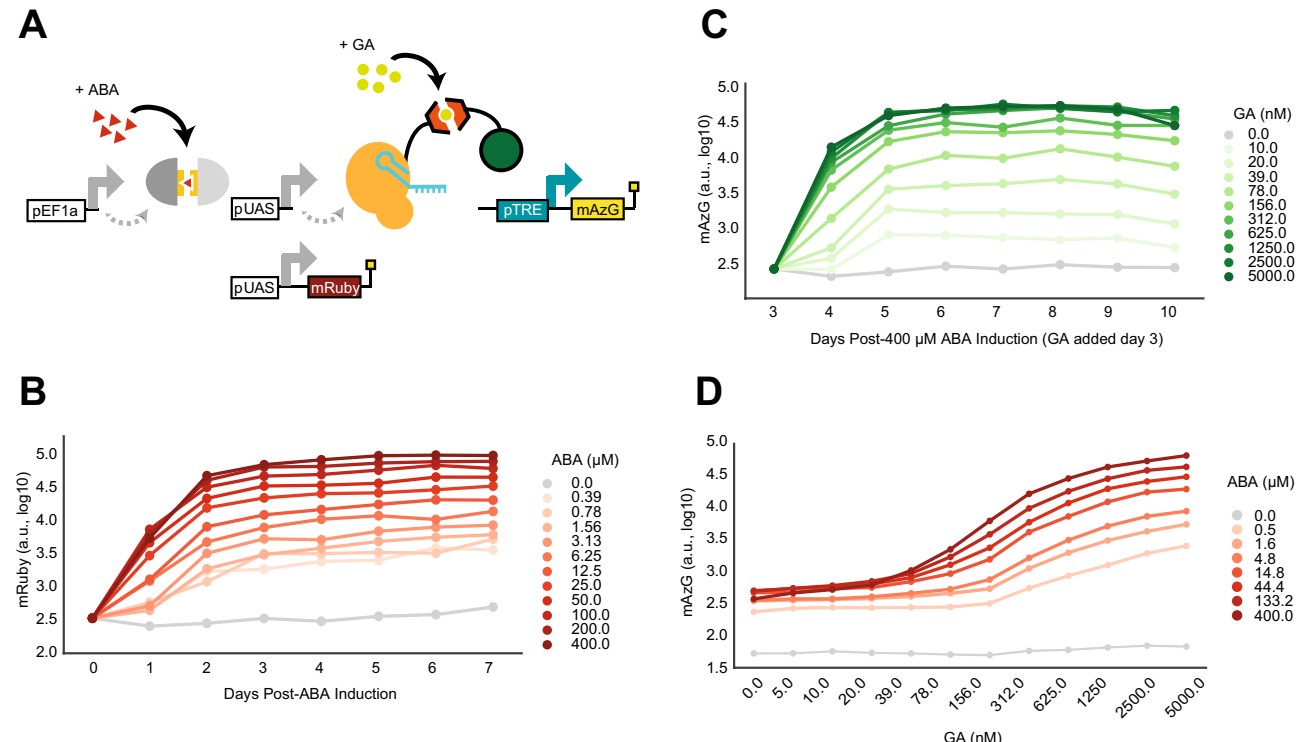

**Fig. 1 TuNR reduces basal leakiness, amplifies fold change, and expands accessible dynamic range relative to single inducible activator for transgene expression. A** Diagram of TuNR circuit composed of a constitutively expressed Abscisic Acid (ABA)-inducible split system consisting of Gal4 and VP-16. This inducible system drives the expression of mRuby as a reporter and a Gibberellic Acid (GA)-induced split system consisting of dCas9 and VPR. Addition of GA and constitutive expression of the guide RNA (gRNA, not pictured) targeted to the Tetracycline Responsive Element (pTRE) drives mAzamiGreen expression. **B** Expression of mRuby from TuNR induced with increasing concentrations of ABA over 7 days. Media was replenished every 24 h. Trace displays the mean of two independent clones. **C** Time-dependent mAzamiGreen expression. Cells were first induced with 400 μM of ABA for 3 days, at which increasing concentrations of GA were added. Measurements were carried out over the following 7 days, while keeping ABA and GA concentration constant through daily replenishment. Trace displays the mean of two independent clones. **D** Quantification of mAzamiGreen at steady state as a function of GA (x-axis) and ABA (shades of red). Data were collected on Day 6 after addition of ABA, and on Day 3 after addition of GA.

ABA for 3 days and then titrating the amount of GA. Similar to the first node characterization, we measured mAzamiGreen expression and replenished media every 24 h over 7 days. As expected, we observed proportional induction of the second inducible node, as mAzamiGreen levels reached steady state by day 6 for all dosages (Fig. 1C). In each of these experiments, we observed an ~100-fold induction for each respective node, consistent with previous reports[35]. Cells that were not exposed to either ABA or GA had nearly tenfold lower mAzamiGreen expression than ABA-primed cells due to the lack of basal dimerization from the split GA recruitment domains (Supplementary Fig. 1D). Importantly, induction of TuNR with ABA and GA showed that mRuby expression, which reports on the activity of the first node, responds uniquely to ABA, and not to GA, confirming that these two small molecules have little cross-reactivity (Supplementary Fig. 1E).

To explore a more comprehensive range of expression for mAzamiGreen in response to simultaneous ABA and GA induction, we primed the cells with a dose–response of ABA for 3 days and, while continuing ABA induction, titrated induction with GA and measured the expression of mAzamiGreen (Fig. 1D). As expected, the absence of both ABA and GA (Supplementary Fig. 2B, top left corner) set the basal expression of mAzamiGreen with the lowest amount possible from TuNR. As an illustration of the benefit of the cascade transcriptional activator arrangement, without ABA, we detected little change in terminal node activation in response to increasing GA (Supplementary Fig. 2B, top row). Conversely, in the absence of GA, TuNR displays sixfold induction

upon addition of ABA, consistent with earlier experiments, suggesting that leakiness emerges from the accumulation of the first node activator (Supplementary Fig. 2B, first column). When both small molecules are present, TuNR induces expression more than either small molecule alone, reaching a maximum mAzami-Green expression of ~1000-fold when both inducers are at their highest concentration. Notably, a transcriptional activator circuit mediated by GA (rows of Supplementary Fig. 2B) achieves ~100-fold induction. As the concentration of ABA increases, so does the basal expression. This reflects a tradeoff between maximum expression and basal leakiness (Fig. 1D). The serial arrangement of the transcriptional activators attenuates this basal leakiness, while achieving a superior maximum fold-change induction when compared to a single-node circuit.

Inducible gene expression systems both in microorganisms[38–40] and mammalian cells[36,37,41,42] have historically suffered from leaky basal expression in the absence of inducer. Our data indicate that the serial topology of TuNR, through the combination of chemically inducible orthogonal recruitment domains, was able to generate a two-input and one-output system with low basal activity with a smooth continuum of expression values. Intuitively, the cascade structure is acting as a coincidence detector in which the output relies on the unlikely simultaneous activation of two transcriptional activators under basal conditions (no small molecule inputs), therefore mediating a low basal activity. However, upon induction with both small molecule activators, output expression is enabled and can be precisely controlled by titrating both independent inputs. We next set out to investigate

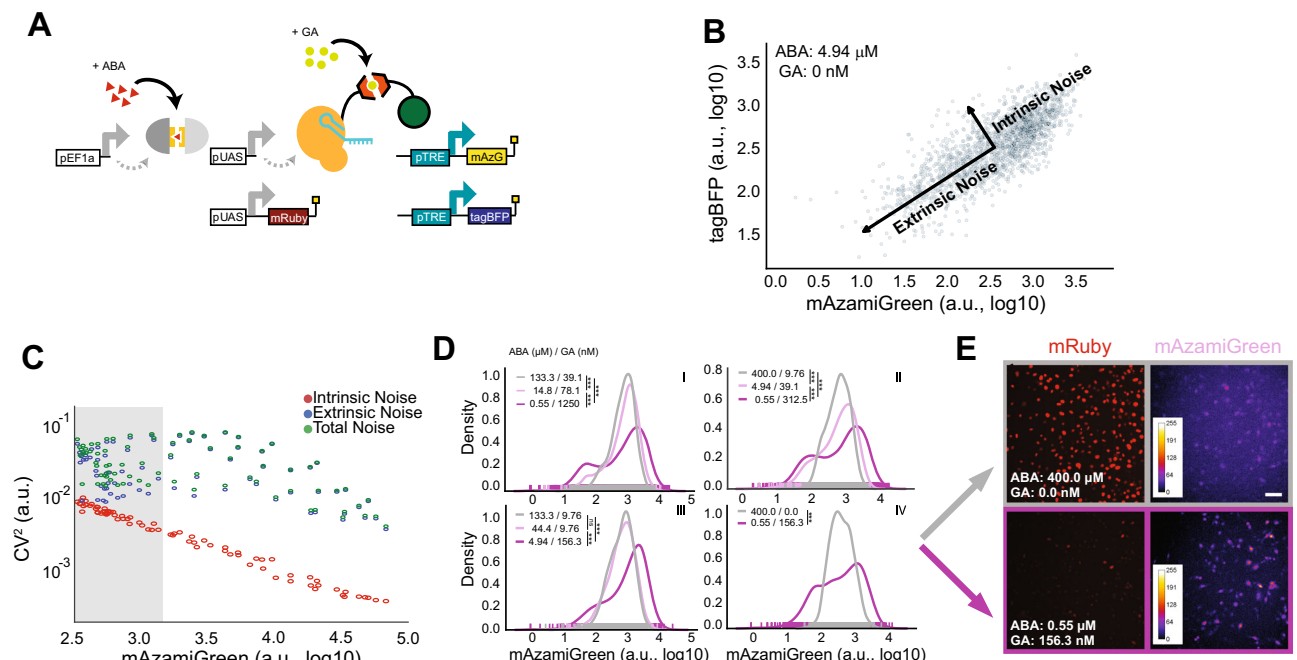

**Fig. 2 TuNR confers independent control of population mean and variance in transgene expression. A** Simplified diagram of TuNR shown in Fig. 1a, with the addition of pTRE driving expression of tagBFP in the same cell to decompose intrinsic and extrinsic noise. **B** Example scatter plot of tagBFP and mAzamiGreen expression from TuNR induced with 4.94 µM ABA and 0 nM GA. Decomposition identifies two major axes of variance: intrinsic variance, which is the spread of points perpendicular to the diagonal; and extrinsic variance, represented by the spread of points along the diagonal. **C** Plot of intrinsic (red), extrinsic (blue), and total (green) noise, quantified using the coefficient of variation or $CV^2$, vs. mean expression of mAzamiGreen of TuNR-containing cells exposed to different combinations of ABA and GA (each ABA/GA combination has an associated intrinsic, extrinsic and total noise value represented by a point). Shaded region represents example area of tunable noise, as demonstrated in **D**. **D** Representative kernel density distributions of mAzamiGreen expression for different combinations of inducer molecules, as identified in **C** (i, ii, iii, and iv), which achieve the same mean expression from one of two independent experiments. ***$P_{adj} <<< 0.05$; two-sided Kolmorgorov–Smirnov test (Bonferroni corrected), NS: not significant. Trace 1 (gray), trace 2 (light purple), and trace 3 (dark purple). (i) 1 : 2, $p = 2.03E − 32$; 1 : 3, $p = 8.51E − 07$; 2 : 3, $p = 1.09E − 18$. (ii) 1 : 2, $p = 4.83E − 12$; 1 : 3, $p = 9.42E − 53$; 2 : 3, $p = 5.48E − 24$. (iii) 1 : 2, NS; 1 : 3, $p = 6.16E − 55$; 2 : 3, $p = 2.36E − 49$. (iv) 1 : 3, $p = 1.17E − 31$. Scale bar, 50 µm. **E** Representative images of mRuby expression and pseudo-colored mAzamiGreen expression from one of two independent clonal replicates corresponding to cells from the populations shown in **D**, isomean IV.

whether cell-to-cell variability might also be modulated in this circuit topology through different combinations of the small molecule inputs.

**Serial topology of TuNR enables independent control of transgene mean expression and noise.** The topology of the TuNR circuit has been shown to allow for the independent control of the mean and variance of the expression of transgenes in yeast[29,34]. To explore the ability of TuNR to control population heterogeneity in the expression of a transgene of interest in mammalian cells, we generated a clonal PC9 cell line that has two identical pTRE promoters that are targeted by the second node and independently drive the expression of both mAzamiGreen and tagBFP (Fig. 2A). We induced TuNR with a two-dimensional dose–response of ABA and GA as previously described, and measured mAzamiGreen and tagBFP expression at steady state. Induction of TuNR with ABA and GA showed similar effects on the mean expression of mAzamiGreen and tagBFP, with both fluorescent proteins displaying correlated, increasing expression with both inducers (Supplementary Fig. 2).

To quantify the total noise for every combination of ABA and GA, we utilized a common noise decomposition strategy to ascertain the extrinsic and intrinsic contributions to the expression noise as shown previously[8]. In this analysis, the correlated expression between the two terminal fluorophores represents the extrinsic noise, or cell-to-cell variability, whereas the uncorrelated

expression is the intrinsic noise, or the cumulative intracellular stochastic effects (Fig. 2B and Supplementary Fig. 3)[43]. Based on prior studies, we hypothesized that due to the serial topology of TuNR, different combinations of ABA and GA could achieve the same mean expression, but with different noise values. To quantify the noise in the system we used the coefficient of variation ($CV^2$). Consistent with the notion that stochastic effects due to counting noise diminish with increasing mean, we observed a strong anti-correlated relationship between intrinsic noise and mean expression (Fig. 2C). Contrary to the intrinsic noise trend, we observed that different combinations of ABA and GA achieved the same mean with different extrinsic noise values (Fig. 2C). We further investigated these "isomean" distributions, and a common pattern emerged: cells exposed to the lower amount of ABA and higher amount of GA had histogram distributions with greater $CV^2$, suggestive of more cell-to-cell variability, than cells exposed to high amounts of ABA and lower of GA (Fig. 2D and Supplementary Fig. 4). The multiplier effect between serially arranged transcriptional activators, which has been previously demonstrated to achieve noise modulation, maintains its efficacy in this more complex eukaryotic system[10,34]. The ability of this circuit topology to produce combinations of inducer molecules that achieve the same mean output but with different variances was predicted by an earlier computation model[34]. Specifically, the model predicted that different output noise levels could be achieved, with high output variance for low levels of ABA and high levels of the GA and low variance for high levels of ABA and

low levels of GA. This observation was consistent across both fluorophores and another clonal cell line of the circuit that likely has a different number of circuit integrations, suggesting that the circuit topology of TuNR is driving this behavior (Supplementary Fig. 5). To visualize the heterogeneity, we imaged the mRuby and mAzamiGreen expression in two representative isomean wells (Fig. 2E). Images of these cells further reinforce that a high concentration of ABA (high expression of mRuby, top row) combined with low concentration of GA gives rise to a tighter distribution that displays a more uniform, average mAzamiGreen expression among cells. Conversely, a lower concentration of ABA (low expression of mRuby, bottom row) with higher amounts of GA lead to more heterogeneous distributions and the presence of both low- and high-expressing mAzamiGreen cells (Fig. 2E). These data establish TuNR as a synthetic circuit that can be deployed to decouple mean expression and the noise of a transgene of interest.

**TuNR enables independent control of population mean and variance of endogenous genes**. Given the capabilities of dCas9, we next attempted to challenge TuNR by testing its efficacy and modularity against endogenous loci. We chose the genes *NGFR* and *CXCR4* for the first proof-of-concept, because their encoded proteins are both membrane-bound and could be stained at the surface of live cells with commercial antibodies. In addition, *NGFR* and *CXCR4* have been previously implicated as pro-liferative and metastatic oncogenes, respectively[21,44]. These two genes represent distinct expression paradigms to assess inde-pendent mean and variability control: (i) *NGFR*, which is not expressed in PC9 cells[21]; and (ii) *CXCR4*, which is constitutively transcribed by the parental cell line[45]. By targeting *NGFR*, we sought to test the extent of control in a gene absent active tran-scriptional machinery at its locus. In the case of *CXCR4*, due to its native constitutive expression, TuNR would compete with endogenous transcriptional and translational machinery in reg-ulating protein abundance.

To target the circuit to endogenous loci in a modular manner, we built a modified TuNR "chassis" clonal cell line without a gRNA cassette and confirmed that the first node of the circuit reached steady state with comparable kinetics to the original TuNR (Supplementary Fig. 6A, B). We transduced the cells with lentivirus carrying previously vetted gRNAs that target either the *NGFR* (Fig. 3A) or *CXCR4* (Fig. 3B) promoters with a tagBFP reporter indicative of integration[46]. We then induced cells with ABA and GA at varying concentrations, reproducing the two-dimensional dose–response matrix described earlier, and measured protein levels of *NGFR* and *CXCR4* (Supplementary Fig. 6C, D).

TuNR achieved 7.2-fold mean induction for *NGFR* and 3.4-fold induction for *CXCR4* and (Fig. 3C, D), which are levels comparable to what other systems have achieved with CRISPRa[47,48]. In addition, as observed in modulating mAzami-Green, TuNR showed a negligible effect on basal levels of *NGFR* and *CXCR4* (Fig. 3C, D), demonstrating that TuNR minimally perturbs basal gene expression due to its serial topology.

Furthermore, distributions of protein abundance at different levels of ABA and GA induction showed a nearly three- and two-fold range of tunable extrinsic noise for *NGFR* and *CXCR4*, respectively (Fig. 3E, F). Here again, when comparing distribu-tions with the same mean for both *NGFR* (Fig. 3G, I) and *CXCR4* (Fig. 3H, J), we found that low ABA and high GA gave rise to distributions with a greater $CV^2$, whereas high ABA and low GA reproducibly gave rise to distributions with a lower $CV^2$ of expression in the cell populations (Supplementary Figs. 7 and 8). These results illustrate that TuNR is able to precisely and orthogonally control select genes from their endogenous loci and produce cellular populations with distinct means and variances in a manner consistent with transgene regulation.

**Discussion**
Advances in synthetic biology have enabled novel investigations into fundamental aspects of biology and ushered in a sea change in biotechnology and bioengineering. The main driver of this paradigm shift is the development of new and more precise tools that are modular, robust, and open new lines of questioning. Despite enhancements in the suite of tools available for mam-malian gene regulation, the ability to finely control the mean and variability of gene expression has long-been outstanding, and previous efforts in mammalian expression systems have often convolved these two parameters. To address this need, we developed TuNR, which acts as a versatile and modular tool to effectively decouple control of the mean from the variance in gene expression at steady state. To accomplish this, we arranged two orthogonal, inducible expression systems serially so that by tuning the concentrations of each respective inducer, we can achieve combinations with the same mean expression, yet dif-ferent extrinsic noise properties for targeted genes of interest. From this topology, we demonstrated a dynamic range of nearly 1000-fold inducible transgene expression while reducing basal leakiness 10-fold when compared to a single-node circuit. In addition to this precise control over the mean, this circuit topology enables independent control over the population variability. With the inherent versatility of CRISPR technology, developing a chassis TuNR allows multiple genes to be investi-gated in parallel. To that end, the precise control of gene expression mean and variability was not limited to transgenes, and extended to endogenous loci.

We believe the main contribution of TuNR is in its ability to be a multifaceted tool towards precise gene regulation. Although the induction capabilities of TuNR and other comparable CRISPRa-based systems in activating endogenous gene expression is modest relative to transgenes, we believe that the precise reg-ulation of the distribution of gene expression even within this limited range will be of tremendous value in future investigations. This is largely because the range of noise titration achieved by TuNR seems to be comparable to that of endogenous human promoters[48,49]. Furthermore, the innovation presented by TuNR takes a particular significance given recent findings that suggest that bacteria such as *Bacillus subtilis* have evolved to rarely be capable of independently controlling gene expression mean from variability, leading to a suggestion that similar limitations may exist in mammals[50]. Therefore, a tool such as TuNR that can achieve this decoupling of gene expression and variance presents an opportunity to investigate the costs or opportunities presented by the fact that variability of gene promoters might be inex-tricably chained to a given level of noise, or vice versa.

However, despite the versatility of TuNR, it is likely to be that our ability to achieve relatively small fold changes for endogenous genes as compared to transgenes is related to a lack of clear understanding of enhancer–promoter mechanisms and corrective cellular mechanisms that counteract the action of the synthetic circuit. Understanding these effects will enable synthetic circuits to more robustly drive endogenous gene production. Tentatively, some of the induction discrepancy between endogeneous and transgenes can be bridged by modifying the terminal effector domain with a Sun-tag system, which has demonstrated robust endogenous induction capabilities[48]. Alternatively, using the current iteration of TuNR, one could introduce the com-plementary DNA of a gene of interest under a synthetic promoter (e.g., pTRE) to test whether the induction capabilities recapitulate that of the fluorescent reporters.

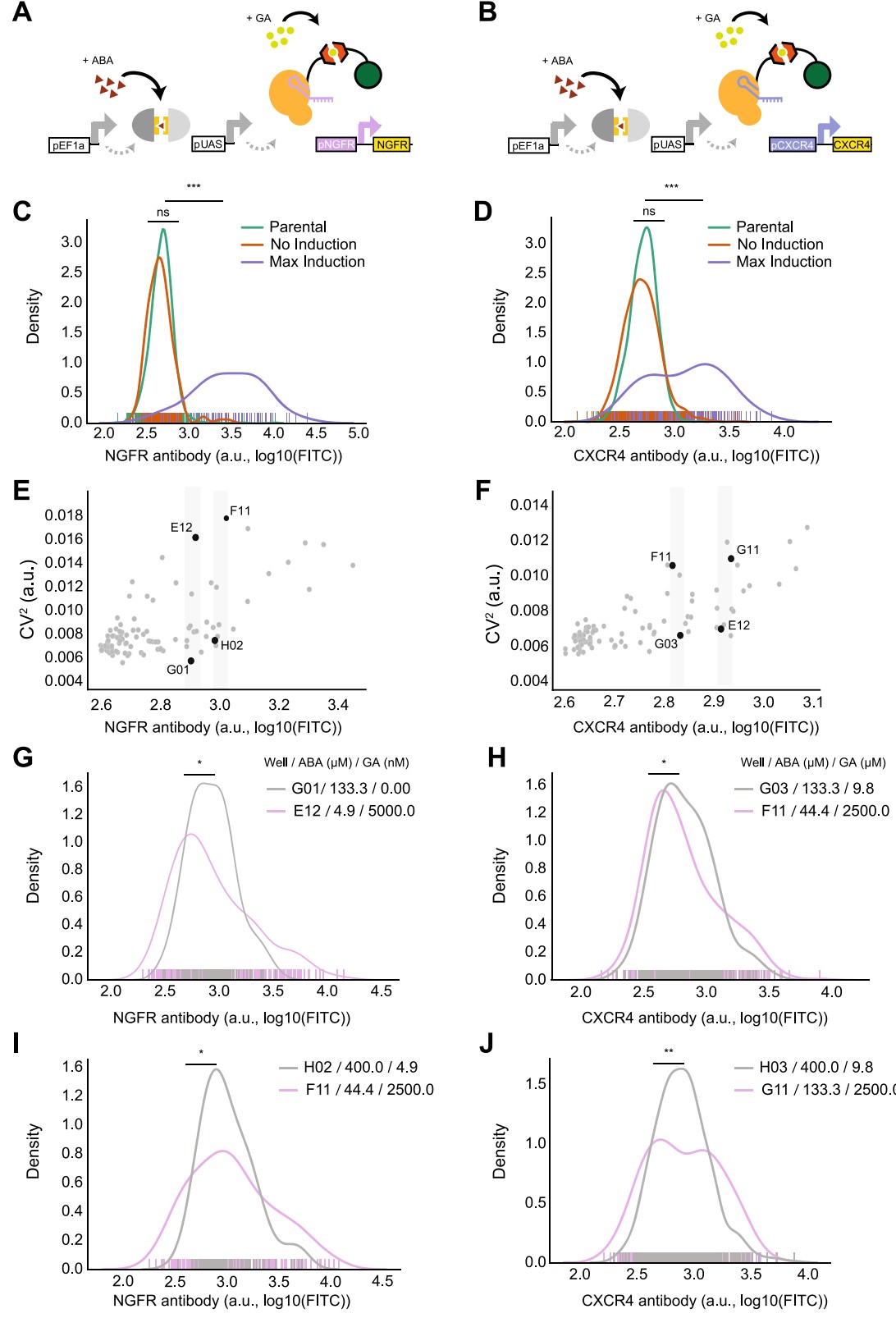

As much has been learned upon the adoption of titratable expression systems vs. unregulated overexpression, we anticipate that granular control over the shape of a gene expression distribution will make similar contributions to the field. Furthermore, as it is well established that genes rarely operate in isolation, by modifying the gRNA cassette, one can conceivably multiplex target modulation in the same controlled manner. Finally, by establishing the TuNR cascade topology as a tractable and versatile system in mammalian cells, it is now possible to attach different effector domains to the circuit's second node to further probe the effects of noisy gene perturbations, not limited to CRISPRa. Overall, the work presented here opens new avenues

**Fig. 3 TuNR enables orthogonal control of population mean and variance of endogenous genes. A** Simplified diagram of TuNR, as shown in Fig. 1A, driving the expression of *NGFR* and **B** *CXCR4* from their endogenous loci. **C** Kernel distributions of *NGFR* expression for no induction (orange) and max induction (purple) of TuNR. Also shown is the *NGFR* distribution in the parental cell line (green). $p = 1.97E − 39$. **D** Kernel distributions of *CXCR4* expression for no induction (orange) and full induction (purple) of TuNR. Also shown is the *CXCR4* distribution in the parental cell line (green). $p = 9.62E − 33$. **E** Coefficient of variation of endogenous *NGFR*, as a function of mean expression, following the activation of TuNR for 6 days. **F** Coefficient of variation of endogenous *CXCR4* as a function of its mean expression, following the activation of the TuNR for 6 days. **G, I** Representative kernel distributions of *NGFR* for different TuNR induction levels that achieve the same mean (isomean) but different variability as identified by gray strips in **E**. Panel **G** $p = 0.02946$; Panel **I** $p = 0.0182297$. **H, J** Representative kernel distributions of *CXCR4* for different TuNR induction levels that achieve the same mean (isomean) but different expression variability, as identified by gray strips in **F**. Panel **H** $p = 0.03467$; Panel **J** $p = 0.0057441$. *Padj < 0.05; **Padj ≪ 0.05; two-sided Kolmorgorov–Smirnov test (Bonferroni corrected), NS: not significant.

to precisely interrogate one of the most fundamental aspects of biological systems: cell-to-cell variability. TuNR is a powerful new tool that will enable genetic perturbations with precise control and thus allow for future studies to answer expression variability- and magnitude-dependent questions.

## Methods

**Plasmid construction**. Plasmids were constructed using a hierarchical DNA assembly method as described previously[51,52]. In short, a previously vetted library of circuit components were assembled into transcriptional units following a BsaI Golden Gate (GG) reaction of constituent plasmid parts. Following bacterial transformation and subsequent miniprep, transcriptional units were verified with a diagnostic restriction digest. In the final step, constituent transcriptional unit plasmids were assembled in a BsmBI GG reaction, transformed, miniprepped, and verified with a diagnostic restriction digest. All proteins had internal BsaI and BsmBI restriction sites removed prior to cloning. Proteins were introduced into the cloning system either by PCR or gene block (IDT), and were then assembled into transcriptional units. Transcriptional units were then assembled into final mutli-transcriptional unit (multi-TU) destination vectors to facilitate delivery to cells as described in the main text. To increase efficiency of integration, TuNR construct was split into two plasmids encoding full circuit. A third plasmid encoded pTRE driving mAzamiGreen and tagBFP, along with a mu6 cassette expressing a gRNA targeted to the pTRE promoter (Supplementary Table 1). Primers used to generate gRNAs can be found in Supplementary Table 2. All plasmids and cell lines will be available at Addgene or upon request to the corresponding author.

**Bacterial cell culture**. Commercial MachI and XL10 strains (QB3 MacroLab) were used to transform plasmid vectors. A typical transformation mixture consists of 2 μL of the GG reaction product, 48 μL bacteria, incubated on ice for 30 min, heat shocked at 42 °C for 1 min, recovered on ice for 5 min, reaction mixture plated onto selective agar, and incubated overnight at 37 °C. In the case of multi-TU transformations, cells recovered in LB media for 30 min after heat shock at 37 °C before plating reaction onto kanamycin-selective agar plates. Cells were cultured in antibiotic concentrations of 100 μg/mL chloramphenicol (part domestication), 25 μg/mL carbenicillin (transcriptional unit), and 100 μg/mL kanamycin (multi-TU).

**Mammalian cell culture**. PC9 cells were maintained in RPMI media (Thermo) supplemented with 10% fetal bovine serum (UCSF Cell Culture Facility), 1% Glutamine (Gibco), and 1% Anti-Anti (Gibco). Cells were passaged every other day and maintained at 37 °C with 5% $CO_2$. For flow cytometry, cells were seeded at 1500 cells/well in a 96-well flat-bottom plate (Corning) and allowed to adhere overnight. Cells were induced with ABA and fresh media with ABA drug was replenished every 24 h for 3 days. After 72 h of ABA induction upon which steady-state has been reached, cells were induced with ABA and GA for 3 additional days.

**Cell line generation**. Cell lines used in Figs. 1 and 2 were generated by co-transfecting parts one, two, and three of the TuNR circuit in equimolar amounts in addition to PiggyBac Transposase (pCMV-hyPBase) using Lipofectamine 3000 (Thermo) according to the manufacturer's instructions. TuNR "chassis" cell lines were generated as described above except by omitting part 3. All cell lines were clonally expanded from a single cell and verified to express circuit components by fluorophore proxy.

**Lentiviral production**. Lentivirus particles were generated as described previously[51]. In short, LX-HEK293T cells were seeded at ~50% confluency in a six-well plate and, the following day, were transfected with lentiviral vector of interest alongside packaging plasmids (pCMV-dR8.91 and pCMV-VSV-G) using Lipofectamine 3000 (Thermo) according to the manufacturer's instructions. After 72 h, the supernatant was filtered through a 0.45 μm filter and added to PC9 cells in standard growth media supplemented with 4 μg/mL of polybrene (SCBT

sc-134220) and centrifuged at $800 \times g$ for 30 min. After 24 h, media was exchanged for fresh media and assessed for selective marker expression after 72 h.

**Flow cytometry**. Flow cytometry was performed using a LSR Fortessa (BD) with a four laser configuration (488, 635, 355, 405 nm). mAzamiGreen (excitation at 488 nm, emission at 530 nm), mRuby (excitation at 561 nm, emission between 610 and 620 nm), tagBFP (excitation at 355 nm, emission at 450 nm), and iRFP713 (excitation at 690 nm, emission at 713 nm) fluorescence levels were recorded for at least 10,000 events. To isolate single-cell clones, cultures were fully induced with ABA and GA, and single cells were isolated using a FACS Aria II (BD) into a 96-well flat-bottom plate based upon iRFP713, mRuby, and mAzamiGreen expression. Clonal cells were again screened for fluorescence activation after three weeks. For all experiments, cells were gated based on iRFP713 expression (presence of circuit). Data were processed using FlowCytometryTools v0.5.0 and SciPy v1.1.0 package in Python 2.7.

**Microscopy**. Images were collected on a Nikon Ti Inverted Widefield Epi-fluorescence microscope with a mercury lamp to illuminate mAzamiGreen (excitation at 488 nm, emission at 530 nm) and mRuby (excitation at 561 nm, emission between 610 and 620 nm). Imaging was performed in a temperature and atmosphere controlled chamber collected through a 10× air objective with a 200 ms exposure time. Image histograms were normalized and pseudo-colored in Fiji (ImageJ).

**Drug compounds**. ABA (Sigma) and GA (Santa Cruz Biotechnology) were prepared as individual 4000× stocks (1.6 M and 20 mM, respectively) and used to generate 96-well plates of mixed stock plates described below.

**Drug stock preparation**. Using an automated liquid handler (Labcyte Echo), drug stocks were prepared as 96-well plates (BioRad) at stock concentrations used for replicate experiments. Stock volumes were held constant with dimethyl sulfoxide and plates were stored at −20 °C.

**Immunostaining**. Cells were dissociated non-enzymatically using Versene Solution (Gibco™) at 37 °C for 20 min. The cells were then quenched in media and resuspended with wash buffer (10% fetal bovine serum in Dulbecco's phosphate buffered saline) and transferred to a V-bottom 96-well plate. The cells were pelleted by centrifugation at $400 \times g$ for 4 min. The plate was rapidly decanted and resuspended with 50 μL of corresponding antibody (1 : 400 in wash buffer) and incubated at room temperature in the dark for 1 h. The cells were then washed with 100 μL of wash buffer and resuspended in 100 μL of wash buffer for immediate analysis by flow cytometry. Antibodies against CXCR4 were purchased from ThermoFisher (#53-9991-80) and antibodies against NGFR were purchased from BioLegend (#345104). CXCR4 antibody and isotype were used at 1.25 μg/mL, and NGFR antibody and isotype were used at 1.25 μg/mL.

**Data processing and statistical analysis**. Statistical and data analysis was executed using custom-written Python scripts.

*Calculating CV²*. All histogram distributions were acquired at the same time once the cells in every experiment were at steady state (exception in Fig. 1B, C). The $CV^2$ was calculated as: $\left(\frac{SD}{\mu}\right)^2$ of each population of cells at steady state, representing the variability across cells at that time.

**Reporting summary**. Further information on research design is available in the Nature Research Reporting Summary linked to this article.

## Data availability
Source data are provided with this paper.

## Code availability

All custom code is provided for this paper and is available at https://ucsf.box.com/s/he29gcnt6igblwo56jvtgg4p34k2gzrg.

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

## Acknowledgements

We thank members of the El-Samad lab for helpful guidance, feedback, and discussion. We also thank Jason Town (UCSF), Ricardo Almeida (UCSF), Xiaoxiao (Vany) Sun (UCSF), Qiu Chang Wu (Harvard), and Zoë Steier (UC Berkeley) for early assistance and discussions. We thank the Steven Altschuler and Lani Wu labs for the gift of parental PC9 cells and the Wendell Lim lab for LX-HEK293T cells. H.E.-S is an investigator in the Chan Zuckerberg Biohub and this work was supported by the CZ-Biohub gift. This work was also supported by the National Science Foundation grant DBI-1548297 and National Science Foundation through grant NSF-MCB 1715108 awarded to H.E.-S, and the National Defense Science & Engineering Graduate (NDSEG) Fellowship awarded to A.R.B.

## Author contributions

A.R.B., J.P.F., and H.E.-S. conceived the study. A.R.B., J.P.F., and J.E.P. designed and performed all experiments, and collected data. All authors interpreted the results and wrote the manuscript.

## Competing interests

The authors declare no competing interests.
