## [Peer Review File · Nature Communications]

REVIEWER COMMENTS

Reviewer #1 (Remarks to the Author):

This paper by Bonny, A.R. et al reports a new tool, called TuNR, for independently modulating the output mean and variability of gene expression in mammalian systems. To do this, the researchers serially coupled two transcriptional activators, which are activated by two orthogonal small molecule inputs (ABA and GA). The researchers first demonstrate the ability of this system to tune mean output of an exogenous gene. They then explore the ability of this system to control the variance of exogenous, endogenous but unexpressed, and endogenously and constitutively transcribed genes. This paper is well written, and a tool to independently modulating the mean and variability of gene expression, as developed in this work, could be broadly useful. While previous papers have demonstrated similar concepts, the authors provide advances by demonstrating their technique in mammalian cells and targeting endogenous promoters.

However, the individual components that make up TuNR, as well as its general approach (serial arrangement), are not novel, and have all been described previously. The control TuNR achieves over variability is also at the same level as in earlier work (spanning 2 - 3 fold). Due to the limited novelty and scope of this work, it is important for the authors to provide a thorough characterization of their system, so that it may be useful for readers. This, and several important concerns about the data collection and analysis, require significant revision.

Major points

1. Because the authors present TuNR as a tool for independently modulating the mean and noise of gene expression, the authors should provide systematic demonstration of this point. The authors do provide thorough analysis of how mean gene expression changes as a function of inducer concentrations (Fig. 1D and E). However, a comparable analysis on noise is lacking. For instance, in Fig. 3G and H, what ABA and GA concentrations led to these values? Do they represent the full set of possible mean and variability values achievable with TuNR? A more thorough analysis would be beneficial.
2. Related to the earlier point, the authors should include descriptive mathematical modeling relating the dosages of input small molecules to the output mean and variability. This would improve both understanding of this system and its usability.
3. The lack of robust statistical analysis of the different noise levels in endogenous gene expression raises questions of reproducibility. The analysis in Supplementary Figure 7, which shows positive correlation between the CV² of two replicates, is insufficient. How much does the "target" variability vary over repeated experiments? This would seem to be an important question to answer for a tool that is intended to control variability.
4. The data in Fig. 1D and Fig. 1E seem to be identical and should not be presented separate figures. The graph in Fig. 1E appears preferable, as it more readily allows quantitative comparisons.
5. Fig. 2E is an image representing two cell populations with equal means and different variability of mAzamiGreen fluorescent intensity. However, the mAzamiGreen image in the lower panel appears to contain many more bright cells, and fewer dim cells, than in the upper panel, calling to question how the conclusion of isomean fluorescent intensities came about. Since fluorescent quantification is the main method of data collection and analysis in the paper, the methodology should be rigorous. The authors should consider amending their methodology to preclude the possibility that their analysis has been confounded by fluorescent bleed-through by: 1) including the use of compensation beads in all flow cytometry analysis with more than one fluorophore, and/or 2) repeating the experiment in Fig. 2 with an engineered cell line without the mRuby reporter, so that only a single color is extant.
6. In the Discussion, the authors write that "extrinsic noise" can be titrated using TuNR. Although authors demonstrate this with fluorescent proteins by decoupling intrinsic from extrinsic noise, it is unclear if a comparable analysis was performed on endogenous genes. This should be clarified and/or appropriate analysis and experiments performed.

7. What is the advantage of targeting the promoters of endogenous genes (as in Fig. 3), rather than introducing the same TuNR chassis in Fig. 1-2 except with the fluorescent proteins replaced with endogenous proteins? It is good that the authors tested the compatibility with TuNR with endogenous promoters, but introducing the chassis with the same promoters as in Fig. 1-2 may recoup some of the substantive induction capabilities of the original chassis. The authors should explore this point.

Minor points

1. The schematic in Fig. 1A should incorporate more labels to improve clarity. For instance, it is unclear, both from the schematic and the text, how mRuby was utilized as a marker for the output of the first circuit.
2. In Fig. 1B, C and E, the colors of the low induction data points are difficult to see and should be changed.
3. The bottommost legend item in Fig. 1E is unlabeled (dark red dot).
4. The graph in Fig. 3C is difficult to read due to overlapping data points of intrinsic and extrinsic noise. Using clear data points with solid outlines for the data series in front may help.
5. The relation between Fig. 3E, and 3G and I; and Fig. 3F, and 3H and J should be clarified in the figure and/or figure captions.
6. What is the y-axis "density" in all the histogram plots? It cannot be fraction of the whole, since values often exceed 1.0. This should be clarified.

Reviewer #2 (Remarks to the Author):

The authors propose a synthetic gene circuit composed by a transcriptional cascade that achieves decoupled control of gene expression response mean and variability via two independent chemical inducers. The main contribution of the work, which essentially borrows the circuit from previous work in yeast, is to engineer the scheme in human cells and to herein demonstrate independent mean and variability control. The claimed impact of the work is the possibility to study effects of noise along with those of average gene expression for arbitrary genes in the mammalian cells.

The paper is well written and clear, except for a few points (see below). In general, it is convincing about the quality of the work. In terms of contribution, the authors propose a novel tool for analysis of noise in human cells. The contribution is timely and original to this reviewer's knowledge. Despite its interest though, there are limitations, as follows.

The synthetic biology design per se is not novel given its previous introduction in yeast. The implementation in human cells certainly presents new challenges, which may themselves bring about advancements in synthetic biology, but limited emphasis is placed on this. Also authors do not contribute any specific study of noise or its effects in human cells. Therefore, as is, the contribution is providing a novel tool but its impact hangs upon arguments based on the literature, not on novel results or any application presented in the paper where the contributed solution plays a fundamental role.

From the technical viewpoint, I have the following remarks:

- The control of the transcriptional cascade is operated via chemical inducers. Compared to current optogenetics solutions, reversibility and speed of the control is questionable. In particular, this seems to exclude dynamical applications (indeed the authors only look at steady states of rather slow expression dynamics). Is this the intended use of the circuit, and is this sufficient to tackle the applications that authors have in mind as a motivation of this work ?

- A key prerequisite brought forward by the authors is that noise control via external inducers should only target the intended gene expression variable, without "side effects". While the authors provide evidence that the observed noise control is indeed a result of their circuit, it does not seem to me that in the paper they provide evidence to fulfill the prerequisite stated by themselves, that is, that the circuit only affects its own gene expression noise. This puts into question the "orthogonality" of the proposed module to the host cell dynamics.

- The strength of noise achieved is rather limited, with a CV₂ ranging below 10⁻¹ (in fact, below 0.02). Despite the possibility to tune this strength, I wonder if such noise level is at all sufficient to study the effects of noise.

- Authors should be more explicit since the beginning that what can be controlled independently of the mean is the extrinsic noise, while (as they acknowledge) the intrinsic noise level is purely determined by molecule count. In this respect, given the double-reporter assessment technique (Figure 2B and relevant reference 8), it is unclear if the quantified extrinsic noise corresponds to intracellular fluctuations or to cell-to-cell variability. Unfortunately, the authors do not provide any information on how they compute the CV (over time? across cells?): Section "Data Processing and Statistical Analysis" of "Materials and Methods" should provide detailed mathematical information for this quantification

- I could not check the data processing code via the link provided by the authors. I could find a readme.md but no more. The readme.md indicates a link where to find the data hosted on a different server. It is not clear to me whether the code should be found within those data folders (I could only find some python notebooks in some sub-sub-folder), but anyway the authors should make the access to the code clearer, or add the necessary code if missing.

Further minor remarks are as follows:

- Page 3: "However, this strategy [...] non-trivial barriers to implementation in mammalian systems.". What barriers? Reference and/or better discussion needed. References should also be given/reported here for the following statements in the rest of the same paragraph.

- Fig 1 D and E give pretty much the same information to my understanding, why having both? Note that confidence intervals (which would motivate Fig 1 E) are not visible (same in other figures). Color coding (here and in other figures) is also a bit deceptive (red curves to show green reporter).

Also, confidence intervals are computed over two replicates, which is the bare minimum: This feels a bit too few to have a credible interval.. better would be to simply show the two replicates

- Page 6, "Importantly, [...] confirming that these two small molecules have little cross-reactivity": This argument is not very clear. In view eg of Fig 1 D, it seems to me that GA does not influence expression of the first gene, but ABA does influence expression (leakiness?) of the second gene. So their effects are not orthogonal. How to interpret this in terms of "cross-reactivity" I am not sure, authors should sort this out.

- Page 7, "The serial arrangement [achieves] a superior maximum fold-change induction when compared to a single-node circuit." : Evidence for this? Reference ?

- Page 8, discussion of Fig 2C (around line 176): Looking at the figure, it is not clear what combinations of GA and ABA correspond to the points in the cloud corresponding to extrinsic noise. It seems that for different mean values, noise is simply erratic, that is, uncontrollable. As a result, the discussion in the paragraph (notably the sentence "Contrary to the intrinsic noise trend [...] same mean with different extrinsic noise values") is confusing. That the level of noise can be controlled at a given value of the mean for suitable choices of GA and ABA only becomes clear later. Authors should prevent this source of confusion.

- Fig. 2D: This plot shows that variability also takes the form of bimodality, but no comment is made on this. Authors should elaborate.

- Page 9, "The multiplier effect between serially-arranged transcriptional activators, [...]". Couldn't you make a simple stochastic model of this and analyze it / compare with data ? This seems well within reach and would strengthen your analysis

- Page 9 line 185-187: I have troubles understanding this sentence. Also, the referenced Supp Fig 5 is hard to understand for me: What do dots correspond to ? Different levels of ABA and GA ? Is this figure trying to show repeatability of mean and noise levels?

- Fig 3: labels of some horizontal axes seem missing

- Page 10/11, lines 225-227, "we found that [...] gave rise to wider distributions, while [...] gave rise to narrower distributions": Distributions look different but no statistical evidence of it is provided. Author should support their arguments with statistical tests for difference in distributions and for difference in variances (smaller/larger variance tests), with p-values conveying the strength of the evidence

Reviewer #1 (Remarks to the Author):

We thank the reviewer for their encouraging words and for recognizing that our paper advances the field. We wholeheartedly agree that *“a tool to independently modulating the mean and variability of gene expression, as developed in this work, could be broadly useful. While previous papers have demonstrated similar concepts, the authors provide advances by demonstrating their technique in mammalian cells and targeting endogenous promoters.”*

We believe that recent manuscripts that appeared in the literature while we were preparing this revision further enhanced the relevance of our results (which the reviewer nicely summarized above). A rather intriguing paper (Deloupy et al., 2020) appeared demonstrating that bacteria have evolved to be rarely capable of independently controlling gene expression mean from variability. We have added a discussion and reference of these results in the revised manuscript. The paper speculates that this might also be true for mammalian cells. Irrespective, the fact that TuNR can achieve decoupling of gene expression and variability presents an opportunity to investigate the costs and opportunities presented by the fact that variability of gene promoters might be inextricably chained to a given level of noise, or vice versa.

The reviewer brings up important points, specifically with respect to “ a thorough characterization of their system”, which we address below to the great benefit of the paper.

Major points

1. Because the authors present TuNR as a tool for independently modulating the mean and noise of gene expression, the authors should provide systematic demonstration of this point. The authors do provide thorough analysis of how mean gene expression changes as a function of inducer concentrations (Fig. 1D and E). However, a comparable analysis on noise is lacking. For instance, in Fig. 3G and H, what ABA and GA concentrations led to these values? Do they represent the full set of possible mean and variability values achievable with TuNR? A more thorough analysis would be beneficial.

We have amended the figures to be more clear in showing which inducer concentrations lead to which distributions. The concentrations of inducers leading to the histograms in Fig. 3G, H, J, and J are denoted at the top right of each respective panel. We have included language in the manuscript that addresses the regimes where the greatest modulation of noise can be found.

2. Related to the earlier point, the authors should include descriptive mathematical modeling relating the dosages of input small molecules to the output mean and variability. This would improve both understanding of this system and its usability.

We agree with the reviewer that mathematical modeling is useful for building intuition about the results. Our previous work in yeast in (Aranda-Diaz et al., 2016) has presented such a model for

the circuit, explaining its behavior as a function of topology, and extracting the principle of how a low input in the first node of the cascade provides a noisy input into the second, and therefore generally gets amplified by a high input given to the second TF. The same model and the same principles still apply here. We have added a few sentences explaining this concept, with more prominent reference to the paper detailing the model on which this work is based.

3. The lack of robust statistical analysis of the different noise levels in endogenous gene expression raises questions of reproducibility. The analysis in Supplementary Figure 7, which shows positive correlation between the CV2 of two replicates, is insufficient. How much does the “target” variability vary over repeated experiments? This would seem to be an important question to answer for a tool that is intended to control variability.

We thank the reviewers for the suggestion of added rigor to our analyses. To address the aforementioned concerns, we have provided an overlay of the distributions corresponding to the two biological replicates of the 96 different combinations of Gibberellic Acid (GA) and Abscisic Acid (ABA). With each terminal fluorophore (mAzamiGreen and tagBFP), the adjusted p-values were obtained using Kolmogorov–Smirnov multiple hypothesis test (Bonferroni correction). Given the statistical reproducibility of TuNR, we thank the reviewer for this suggestion that substantiates the utility of this tool. We show the overlay of the two clone PDFs and CDFs below, highlighting in grey the cases where the distributions did not pass the statistical test with an adjusted p-value of 0.05. We added to the manuscript an additional supplementary figure (Figure S8) where these analyses are also shown.

Clone A10, mAzamiGreen distributions

Clone A10, mAzamiGreen cumulative density functions (grey, padj < 0.05)

Clone A10, tagBFP distributions

Clone A10, tagBFP cumulative density functions (grey, padj < 0.05)

Clone E9, mAzamiGreen distributions

Clone E9, mAzamiGreen cumulative density functions (grey, padj < 0.05)

Clone E9, tagBFP distributions

Clone E9, tagBFP cumulative density functions (grey, padj < 0.05)

CXCR4, FITC distributions

CXCR4, FITC cumulative density functions (grey, $p_{adj} < 0.05$)

NGFR, FITC distributions

NGFR, FITC cumulative density functions (grey, $p_{adj} < 0.05$)

4. The data in Fig. 1D and Fig. 1E seem to be identical and should not be presented separate figures. The graph in Fig. 1E appears preferable, as it more readily allows quantitative comparisons.

We thank the reviewer for suggesting removal of this redundant information. We have moved the heatmap in panel D to the supplemental materials.

5. Fig. 2E is an image representing two cell populations with equal means and different variability of mAzamiGreen fluorescent intensity. However, the mAzamiGreen image in the lower panel appears to contain many more bright cells, and fewer dim cells, than in the upper panel, calling to question how the conclusion of isomean fluorescent intensities came about. Since fluorescent quantification is the main method of data collection and analysis in the paper, the methodology should be rigorous. The authors should consider amending their methodology to preclude the possibility that their analysis has been confounded by fluorescent bleed-through by: 1) including the use of compensation beads in all flow cytometry analysis with more than one fluorophore, and/or 2) repeating the experiment in Fig. 2 with an engineered cell line without the mRuby reporter, so that only a single color is extant.

We thank the reviewer for bringing attention to the potential issue of bleed-through. To address this concern, we generated a clonal cell line containing the TuNR circuit that does not contain mAzamiGreen, but does produce mRuby in response to ABA. Below are 12 doses of

ABA ranging from 0-400 μ M, where at the highest dose (Well C12), we observe a very weak correlation of 0.3 between the red (x-axis) and green (y-axis) channels, and at other concentrations little to no correlation. Additionally, bleed-through typically occurs when a fluorophore with a lower emission excites a fluorophore with its excitation in the same range. The example images, where each row corresponds to the same cells, show that the opposite is true: the prevalence of high-intensity cells in either channel does not correlate with the opposite channel. Therefore, we argue that bleed-through, while it may be minimal at the highest induction of mRuby, does not significantly alter the efficacy of the tool as described, nor the interpretation of the data. We submit that the particular microscopy fields that we show as examples might be confusing to the reader because of different background fluorescence (which we are reluctant to touch/change in any way). To avoid this confusion, we replaced these images with different ones that have similar backgrounds and hence can convey the visual conclusions better.

6. In the Discussion, the authors write that “extrinsic noise” can be titrated using TuNR. Although authors demonstrate this with fluorescent proteins by decoupling intrinsic from extrinsic noise, it is unclear if a comparable analysis was performed on endogenous genes. This should be clarified and/or appropriate analysis and experiments performed.

We appreciate the reviewer bringing to our attention the opportunity to clarify the intended use cases for TuNR. We present evidence that TuNR is a tool that can position a target gene at the same mean but different noise. We demonstrate with the endogenous genes NGFR and CXCR4 that this property of TuNR extends to endogenous genes, despite the added complexity in modulating endogenous signalling. However, we do not claim that TuNR can decouple intrinsic noise from extrinsic noise. It would be interesting to be able to modulate intrinsic noise using a comparable system, but current methods include inserting a TATA box into the promoter region (Zoller et al., 2015), among others. Again, this is not a feature we claim TuNR is capable of.

7. What is the advantage of targeting the promoters of endogenous genes (as in Fig. 3), rather than introducing the same TuNR chassis in Fig. 1-2 except with the fluorescent proteins replaced with endogenous proteins? It is good that the authors tested the compatibility with TuNR with endogenous promoters, but introducing the chassis with the same promoters as in Fig. 1-2 may recoup some of the substantive induction capabilities of the original chassis. The authors should explore this point.

The reviewer brings up the excellent suggestion that by introducing an additional copy of the cDNA of a Gene of Interest (GOI), one could potentially match the induction capabilities demonstrated with fluorophores. While this would generate many hypotheses, we contend that this is beyond the scope of the work. We intended TuNR to be a tool where using the same genetic circuit, one could achieve superior or comparable induction capabilities to many current systems, while simultaneously controlling the variance of the distribution. One use-case we were particularly eager to demonstrate was that this circuit could be used to interrogate endogenous signalling networks. Also, by adding another copy of a GOI under an inducible promoter, one changes the basal expression, potentially interrupting cis-regulatory elements that may be integral to understanding the GOI within its native network. We put forth TuNR as a proof-of-concept, where in later iterations one could replace VPR with a Sun-Tag system, which has demonstrated robust induction capabilities (Tannenbaum et al. 2014). This said, this idea makes an excellent point for the discussion, and we added it to the discussion.

Minor points

1. The schematic in Fig. 1A should incorporate more labels to improve clarity. For instance, it is unclear, both from the schematic and the text, how mRuby was utilized as a marker for the output of the first circuit.

We thank the reviewer for this comment to make the figure more clear. We made the fact that the mRuby2 is a proxy reporter for the terminal node transcription more explicit.

2. In Fig. 1B, C and E, the colors of the low induction data points are difficult to see and should be changed.

We thank the reviewer for this comment to make the figure more clear. We have changed the color of the low induction to have a greater contrast against the white background.

3. The bottommost legend item in Fig. 1E is unlabeled (dark red dot).

We thank the reviewer for bringing this typo to our attention. The legend items are now aligned to their concentrations.

4. The graph in Fig. 2C is difficult to read due to overlapping data points of intrinsic and extrinsic noise. Using clear data points with solid outlines for the data series in front may help.

We thank the reviewer for this comment to make the figure more clear. We made the points easier to discern following the reviewer's recommendation.

5. The relation between Fig. 3E, and 3G and I; and Fig. 3F, and 3H and J should be clarified in the figure and/or figure captions.

We thank the reviewer for this comment to make the figure more clear. We made the fact that each grey strip in Fig. 3E, F is referring to a subsequent panel more explicit.

6. What is the y-axis "density" in all the histogram plots? It cannot be fraction of the whole, since values often exceed 1.0. This should be clarified.

We are now more explicit in describing the generation of the distributions. The "density" on the y-axis refers to the kernel density.

Reviewer #2 (Remarks to the Author):

We thank the reviewer for stating that our paper "is well written and clear, except for a few points (see below). In general, it is convincing about the quality of the work. In terms of contribution, the authors propose a novel tool for analysis of noise in human cells. The contribution is timely and original to this reviewer's knowledge. Despite its interest though, there are limitations, as follows."

We believe that newer results that appeared in the literature while we were preparing this revision further enhanced the relevance of our results (which the reviewer nicely summarized above). A rather intriguing paper (Deloupy et al., 2020) appeared demonstrating that bacteria have evolved to be rarely capable of independently controlling gene expression mean from

variability. We have added a discussion and reference of these results in the revised manuscript. The paper speculates that this might also be true for mammalian cells. Irrespective, the fact that TuNR can achieve decoupling of gene expression and variability presents an opportunity to investigate the costs and opportunities presented by the fact that variability of gene promoters might be inextricably chained to a given level of noise, or vice versa.

Reviewer states:

“The synthetic biology design per se is not novel given its previous introduction in yeast. The implementation in human cells certainly presents new challenges, which may themselves bring about advancements in synthetic biology, but limited emphasis is placed on this. Also authors do not contribute any specific study of noise or its effects in human cells. Therefore, as is, the contribution is providing a novel tool but its impact hangs upon arguments based on the literature, not on novel results or any application presented in the paper where the contributed solution plays a fundamental role.”

We agree with the assessment of the reviewer that the novelty of the work is technical, having circumvented the many difficulties that even a known circuit implementation might present in mammalian cells. In that respect, we argue that the main contribution of the work is the circuit working reliably and modularly in the mammalian context.

Technical Remarks:

- The control of the transcriptional cascade is operated via chemical inducers. Compared to current optogenetics solutions, reversibility and speed of the control is questionable. In particular, this seems to exclude dynamical applications (indeed the authors only look at steady states of rather slow expression dynamics). Is this the intended use of the circuit, and is this sufficient to tackle the applications that authors have in mind as a motivation of this work ?

The reviewer raises a reasonable shortcoming in our proposed application of TuNR, in that its use is largely confined to when the expression of the GOI is at steady state because of the delays that are inherent to gene expression. To further clarify some of the possible applications, we have edited the introduction and discussion to focus more on use-cases such as a steady state population of cells that receive a stimulus (e.g. a cancer drug), and using TuNR to modulate expression of a target of that stimulus to varying degrees.

- A key prerequisite brought forward by the authors is that noise control via external inducers should only target the intended gene expression variable, without "side effects". While the authors provide evidence that the observed noise control is indeed a result of their circuit, it does not seem to me that in the paper they provide evidence to fulfill the prerequisite stated by themselves, that is, that the circuit only affects its own gene expression noise. This puts into question the "orthogonality" of the proposed module to the host cell dynamics.

We thank the reviewer for bringing this subtle but imperative point to more clarity. We argue that TuNR, which is composed of protein components from different model organisms (Arabidopsis, S. pyogenes, herpes virus), and is controlled by small molecules isolated from plants, that the effect on host cell dynamics should be minimal. While we do not provide evidence that the state of the cell is unperturbed, we believe that the use of non-mammalian proteins and inducers minimizes off-target effects. These proteins have been utilized in a variety of earlier works (Gao et al., 2016, Park et al., 2019, Liang et al., 2006), and therefore we anticipate that TuNR will possess similar potential caveats. However, since we have not demonstrated this through direct experimentation, we removed this strong language from the manuscript, arguing instead that this is expected given previous work and usage of these systems, but not demonstrated directly in the current work.

- The strength of noise achieved is rather limited, with a CV² ranging below 10⁻¹ (in fact, below 0.02). Despite the possibility to tune this strength, I wonder if such noise level is at all sufficient to study the effects of noise.

We thank the reviewer for bringing this point to our attention. The spread of CV² shows that the range of noise TuNR can control is approximately 10-fold (Fig. 2C) and 2.5 fold (Fig. 3E) for transgenes and endogenous genes, respectively. Previous work (Weingarten-Gabbay et al., 2019) globally characterized endogenous human promoters and their noise (CV²), and found that the range among the 990 promoters characterized was approximately 12-fold. While TuNR does not match that full range, we contend that TuNR is sufficient to inject a precise amount of noise in order to investigate the effects of variability in both transgene and endogenous gene expression. Because of the importance of this point, we added it as an item to our updated discussion in the manuscript with appropriate references.

- Authors should be more explicit since the beginning that what can be controlled independently of the mean is the extrinsic noise, while (as they acknowledge) the intrinsic noise level is purely determined by molecule count. In this respect, given the double-reporter assessment technique (Figure 2B and relevant reference 8), it is unclear if the quantified extrinsic noise corresponds to intracellular fluctuations or to cell-to-cell variability. Unfortunately, the authors do not provide any information on how they compute the CV (over time? across cells?): Section "Data Processing and Statistical Analysis" of "Materials and Methods" should provide detailed mathematical information for this quantification

We appreciate the reviewer bringing to our attention this important point where we could offer more clarity. All noise measurements were collected across cells at steady state. We added a section in the manuscript explicitly detailing these measurements and calculations.

- I could not check the data processing code via the link provided by the authors. I could find a readme.md but no more. The readme.md indicates a link where to find the data hosted on a different server. It is not clear to me whether the code should be found within those data folders (I could only find some python notebooks in some sub-sub-folder), but anyway the authors should make the access to the code clearer, or add the necessary code if missing.

We thank the reviewer for bringing this to our attention, and agree that providing ALL data and ALL CODE ready to run, is essential. We have improved the code to make the reproduction of our data from source files simple for any interested user.

Further minor remarks are as follows:

- Page 3: "However, this strategy [...] non-trivial barriers to implementation in mammalian systems.". What barriers? Reference and/or better discussion needed. References should also be given/reported here for the following statements in the rest of the same paragraph.

*We added references that support the technical difficulty in adapting this circuit from *S. cerevisiae* to mammalian cell culture. For example, the process of attaining and vetting clonal expression of the circuit, the paucity of characterized synthetic expression systems, and modulating endogenous gene expression.*

- Fig 1 D and E give pretty much the same information to my understanding, why having both ? Note that confidence intervals (which would motivate Fig 1 E) are not visible (same in other figures). Color coding (here and in other figures) is also a bit deceptive (red curves to show green reporter).

Also, confidence intervals are computed over two replicates, which is the bare minimum: This feels a bit too few to have a credible interval.. better would be to simply show the two replicates

We have amended the figure to both be more succinct and clear in how we represent the replicate experiments. While we agree that plotting the replicates would be an alternative representation of the data, we worry that nearly 24 plots would clutter the panel to where it is uninterpretable. To address the reviewer's concern, however, we have added the replicates of Figure 1 B and C in Supplemental Figure 1.

- Page 6, "Importantly, [...] confirming that these two small molecules have little cross-reactivity": This argument is not very clear. In view eg of Fig 1 D, it seems to me that GA does not influence expression of the first gene, but ABA does influence expression (leakiness?) of the second gene. So their effects are not orthogonal. How to interpret this in terms of "cross-reactivity" I am not sure, authors should sort this out.

To address the cross-reactivity of the two small molecule inducers used in this work, we included Supplemental Figure 1F which shows that the expression of the first gene is virtually unperturbed by addition of GA at all doses. However, as the reviewer correctly points out, in the circuit, the second node is leaky and therefore addition of ABA even in the absence of GA shows a small effect. This is a fundamental caveat in using expression systems. We therefore amended our statements of orthogonality to be more precise.

- Page 7, "The serial arrangement [achieves] a superior maximum fold-change induction when compared to a single-node circuit." : Evidence for this? Reference ?

We have added references that demonstrate the expression range for a ABA-induced, split Gal4 transcription factor is approximately 100-fold, and a GA-induced dimerization to a SpdCas9 is approximately 100-fold, consistent with our data where one inducer is held at a constant concentration (Gao et al., 2016). Despite this, the induction of the second gene when both components are linked serially is ~1000-fold.

- Page 8, discussion of Fig 2C (around line 176): Looking at the figure, it is not clear what combinations of GA and ABA correspond to the points in the cloud corresponding to extrinsic noise. It seems that for different mean values, noise is simply erratic, that is, uncontrollable. As a result, the discussion in the paragraph (notably the sentence "Contrary to the intrinsic noise trend [...] same mean with different extrinsic noise values") is confusing. That the level of noise can be controlled at a given value of the mean for suitable choices of GA and ABA only becomes clear later. Authors should prevent this source of confusion.

We thank the reviewer for this observation. We amended the manuscript to be more clear that the noise control pertains to a regime of mean expressions.

- Fig. 2D: This plot shows that variability also takes the form of bimodality, but no comment is made on this. Authors should elaborate.

We thank the reviewer for this observation. The bimodality present in mAzamiGreen and tagBFP expression is a by-product of the first gene (mRuby expression) being bimodal as shown below in the histogram distributions. One can observe that bimodality is present in rows 2-4, which also correspond to the bimodal distributions of mAzamiGreen and tagBFP. While beyond the scope of this work, we believe that by changing parameters of the upstream node (e.g. constitutive expression level, binding kinetics, promoter binding sites) one can conceivably achieve unimodal expression and thus suppress the bimodality.

Clone A10, mRuby: rows 2-4 are bimodal in first node and that coincides with bimodality in terminal node expression (mAzamiGreen and tagBFP)

- Page 9, "The multiplier effect between serially-arranged transcriptional activators, [...]". Couldn't you make a simple stochastic model of this and analyze it / compare with data ? This seems well within reach and would strengthen your analysis

In the manuscript we reference a model that demonstrated the decoupling of mean from variability with the serial arrangement (Aranda-Diaz et al.,2016). We have expounded upon the comparison with the output of the model and our TuNR data.

- Page 9 line 185-187: I have troubles understanding this sentence. Also, the referenced Supp Fig 5 is hard to understand for me: What do dots correspond to ? Different levels of ABA and GA ? Is this figure trying to show repeatability of mean and noise levels?

We apologize for the lack of clarity in the text and figure. In Supplementary Figure 5, each point corresponds to a unique concentration of ABA and GA. This figure demonstrates the mean expression and noise reproducibility of each concentration of ABA/GA between biological replicates and co-varying terminal fluorophores (mAzamiGreen and tagBFP).

- Fig 3: labels of some horizontal axes seem missing

We intended to illustrate that each plot in Figure 3 shares the same x-axis since they are all plotted as a function of each respective GOI's mean expression. We will add explicit axes to avoid confusion.

- Page 10/11, lines 225-227, "we found that [...] gave rise to wider distributions, while [...] gave rise to narrower distributions": Distributions look different but no statistical evidence of it is provided. Author should support their arguments with statistical tests for difference in distributions and for difference in variances (smaller/larger variance tests), with p-values conveying the strength of the evidence

We appreciate the reviewer's insights to improve the rigor of our analyses. We have added a supplementary document of p-values for the identified isomean examples to quantify how statistically different distributions can achieve the same mean, while also having a quantitatively different CV^2 for the distributions. The denoted p-values coupled with the CV^2 we believe will allay the reviewer's concerns, and we added this information also to Figure 2 (with asterisk marks () denoting the significance as explained in the figure legend). Additionally, a supplementary .xls file contains all associated p-values.*

Figure 2C, subplot(I)

G05-E06: $padj=8.5098049e-07$

E06-B09: $padj= 1.0855071577909667e-18$

G05-B09: $padj=2.02925297e-32$

Figure 2C, subplot(II)

H03-D05: $padj= 4.833348656080193e-12$

H03-B08: $padj= 9.42135043497469e-53$

D05-B08: $padj= 5.478628456854454e-24$

Figure 2C, subplot(III)

G03-F03: $p_{adj} = 0.41532898837591065$

G03-D07: $p_{adj} = 6.157015555567354e-55$

F03-D07: $p_{adj} = 2.3600235308871853e-49$

Figure 2C, subplot(IV)

H01-B07: $p_{adj} = 1.1749978243687772e-31$

Figure 3G, NGFR

NGFR example isomean 1

padj= 0.029457673994530354

*Figure 3I, NGFR example isomean 2
padj= 0.018229660465172205*

*Figure 3H,
CXCR4 example isomean 1
padj= 0.03466619567938605*

Figure 3J, CXCR4 example isomean
2
padj= 0.0057441402005685704

REVIEWERS' COMMENTS

Reviewer #1 (Remarks to the Author):

The authors have done a good job in addressing the points raised in my original review. I recommend the revised paper for publication in Nature Communications.

Reviewer #2 (Remarks to the Author):

In this new revision of the paper the authors have satisfactorily addressed the issues raised in the previous review round. To me, the main issue remains that of the range of attainable CV2 values. The authors proposed comparison with the interrogation of native core promoters from the human genome as per Weingarten-Gabbay et al (2019). The point made is that despite smaller, the range covered by TuNR remains comparable with the range of observed CV2 values in the reference given. Besides the fold-change, which are relative values, the authors should not overlook the absolute values over which their controlled CV2 ranges, which appear to be way smaller than those in Weingarten-Gabbay et al (2019). Yet, the values of CV2 in the latter are observed over a wider range of mean inductions, whereas here the authors are limited to a narrower range of (endogenous gene) mean induction levels. While the mean induction levels in the two works do not seem directly comparable, the fact that authors explore CV2 over a narrower range of mean induction levels makes the argument acceptable (the more so as the authors now develop the point in the discussion section). As a final remark, I am not sure why the authors pointed at a preprint of Weingarten-Gabbay et al (2019), while a published (adapted) version of the same is on Genome Research at doi: 10.1101/gr.236075.118

Reviewer #1 (Remarks to the Author):

The authors have done a good job in addressing the points raised in my original review. I recommend the revised paper for publication in Nature Communications.

We thank the review for the rigorous review and constructive feedback that improved the manuscript.

Reviewer #2 (Remarks to the Author):

In this new revision of the paper the authors have satisfactorily addressed the issues raised in the previous review round. To me, the main issue remains that of the range of attainable CV² values. The authors proposed comparison with the interrogation of native core promoters from the human genome as per Weingarten-Gabbay et al (2019). The point made is that despite smaller, the range covered by TuNR remains comparable with the range of observed CV² values in the reference given. Besides the fold-change, which are relative values, the authors should not overlook the absolute values over which their controlled CV² ranges, which appear to be way smaller than those in Weingarten-Gabbay et al (2019). Yet, the values of CV² in the latter are observed over a wider range of mean inductions, whereas here the authors are limited to a narrower range of (endogenous gene) mean induction levels. While the mean induction levels in the two works do not seem directly comparable, the fact that authors explore CV² over a narrower range of mean induction levels makes the argument acceptable (the more so as the authors now develop the point in the discussion section). As a final remark, I am not sure why the authors pointed at a preprint of Weingarten-Gabbay et al (2019), while a published (adapted) version of the same is on Genome Research at doi: 10.1101/gr.236075.118

We thank the review for the rigorous review and constructive feedback that improved the manuscript. We propose that future iterations of TuNR will be able to extend the range of CV² modulation with some of the enhancements we propose in the discussion. We have corrected the Weingarten-Gabbay et al. citation to appropriately reference the published version.